# Extreme wind fluctuations: joint statistics, extreme turbulence, and impact on wind turbine loads

Ásta Hannesdóttir, Mark Kelly, and Nikolay Dimitrov

DTU Wind Energy Dept., Technical University of Denmark, Roskilde, Denmark

*Correspondence to:* Ásta Hannesdóttir (astah@dtu.dk)

**Abstract.** For measurements taken over a decade at the coastal Danish site Høvsøre, we find the variance associated with wind speed events from the offshore direction to exceed the prescribed extreme turbulence model (ETM) of the IEC 61400-1 Ed.3 standard for wind turbine safety. The variance of wind velocity fluctuations manifested during these events is not due to extreme turbulence; rather, it is primarily caused by ramp-like increases in wind speed associated with larger-scale meteorological processes. The measurements are both linearly detrended and high-pass filtered in order to investigate how these events—and such commonly-used filtering—affect the estimated 50-year return period of turbulence levels. High-pass filtering the measurements with a cut-off frequency of 1/300 Hz reduces the 50-year turbulence levels below that of IEC ETM class C, where as linear detrending does not. This is seen as the high-pass filtering more effectively removes variance associated with the ramp-like events. The impact of the observed events on a wind turbine are investigated using aeroelastic simulations, that are driven by constrained turbulence simulation fields. Relevant wind turbine component loads from the simulations are compared with the extreme turbulence load case prescribed by the IEC standard. The loads from the event simulations are on average lower for all considered load components, with one exception: Ramp-like events at wind speeds between 8-16 m/s where the wind speed rises to exceed rated wind speed can lead to high thrust on the rotor, resulting in extreme tower base fore-aft loads that exceed the extreme turbulence load case of the IEC standard.

## 1 Introduction

The IEC design standard for wind turbine safety (61400-1 edition 3, IEC, 2005) outlines requirements that, when followed, offer a specific reliability level which can be expected for a wind turbine. The standard prescribes various operational wind turbine load regimes and extreme wind conditions that the wind turbine must be able to withstand during its operational lifetime. So-called design-load cases (DLC's) are described, following these prescribed regimes and conditions. One of the IEC prescriptions is an extreme turbulence model (ETM), which gives the ten-minute standard deviation of wind speed, with a 50-year return period, as a function of ten-minute mean wind speed at hub height. The ETM takes into account the long-term mean wind speed at hub height and is scaled accordingly through the wind speed parameters of the IEC wind turbine classes. The model is prescribed in a design load case (DLC 1.3) for ultimate load calculations on wind turbine components; this DLC is considered to be important in wind turbine design, particularly for the tower and blades (Bak et al., 2013). For the standard

to be effective, it must reflect the expected atmospheric conditions and the extreme events that a wind turbine may be exposed to. Likewise, it is important that DLC 1.3 is representative of observed extreme turbulence conditions.

The IEC standard recommends the uniform-shear spectral turbulence model of Mann (1994, 1998) for generation of three-dimensional turbulent flow, to serve as input to turbine load calculations. Gaussian turbulent velocity component fluctuations are synthesized via the 'Mann-model' spectra, and assumed to be stationary and homogeneous (unless the model is modified, as in de Mare and Mann, 2016). The model requires three input parameters, which have values prescribed by the standard. In Dimitrov et al. (2017) it is shown that the parameters of normal turbulence and extreme turbulence differ, and how these differences influence wind turbine loads. There it is also shown how numerous 10-minute turbulence measurements from the homogeneous land (eastern) sectors exceed the ETM model at the Danish Test Centre for Large Wind Turbines at Høvsøre, indicating that the ETM model is not necessarily conservative.

A further investigation of 10-minute turbulence measurements exceeding the ETM level is needed to identify what kind of flow causes these extreme events and how they influence the estimated turbulence level at a given site. Fluctuations associated with mesoscale meteorological motion can have periods in the range of a minute up to hours (Vincent, 2010). In the shorter end of this range the fluctuations are the main contribution to the 10-minute variance estimate (turbulence level). Short-time mesoscale fluctuations have been reported in connection with e.g. open cellular convection (Vincent et al., 2012), convective rolls (Foster, 2005) and streaks (Foster et al., 2006). The fluctuations are seen in measurements as coherent structures with a ramp-like increase in wind speed (Fesquet et al., 2009). These studies have been made with respect to identification, modelling, forecasting and wind power generation, but they do not consider the impact on wind turbine loads.

In this paper we aim to find and examine events where the 10-minute variance exceeds the ETM level. However here we consider them as non-turbulent events, as they are caused by ramp-like increase in wind speed associated with larger-scale meteorological processes, which may be observed offshore or high above the surface layer. We use measurements from the measurement site Høvsøre, focusing on the western (offshore) sectors. We demonstrate how these events influence the estimate of 10-minute turbulence levels with a 50-year return period. This is done for the raw-, linearly detrended-, and high-pass filtered measurements. The observed events are simulated by incorporating measured time series using a constrained simulation approach, in order to get a realistic representation of the flow involved. The generated wind field realizations are fed to an aero-elastic model (Larsen and Hansen, 2015) of the DTU 10MW reference wind turbine (Bak et al., 2013), to investigate how they affect wind turbine loads. Finally, the load simulations with the observed events are compared to simulations corresponding to DLC 1.3 from the IEC 61400-1 standard.

## 2  Site and measurements

The data analysis and load simulations are based on measurements from the Høvsøre Test Centre for Large Wind Turbines in western Denmark. Located over flat terrain 1.7 km east of the coastline, the site offers low-turbulence, near-coastal wind conditions. The site consists of five wind turbines arranged in a single row along the north-south direction, and multiple measurement masts.

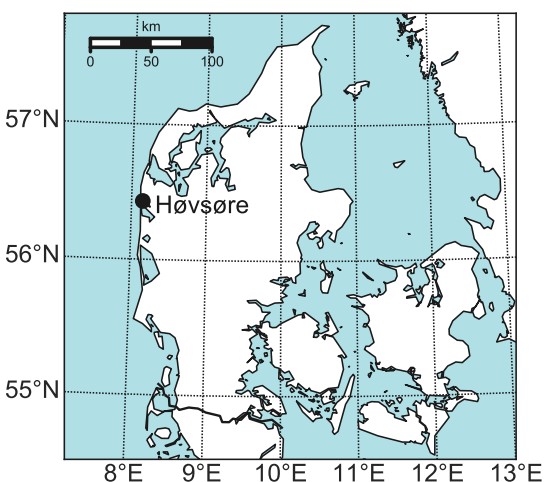 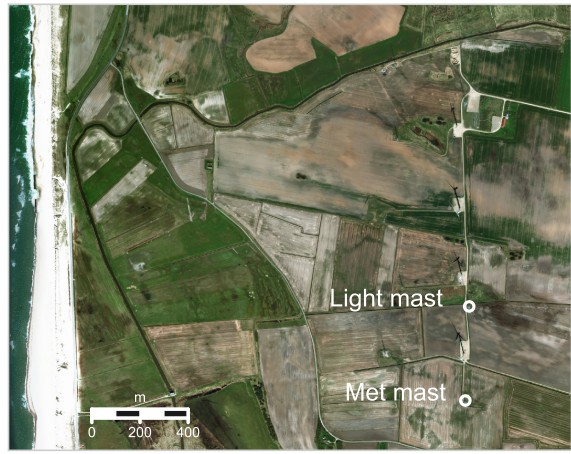

**Figure 1.** *Left:* Map of Denmark showing the location of Høvsøre. *Right:* Overview of the Høvsøre test center showing the position of the met mast and the light mast with white circles.

The primary data source used in this paper is a light mast[1] placed between two of the wind turbines. This mast has cup anemometers and wind vanes at 60 m, 100 m and 160 m heights installed on southward pointing booms. The measurements span a 10-year period, from November 2004 to December 2014 and the recording frequency is 10 Hz. The light-mast data is compared with data from the main Høvsøre meteorological mast, which is located south of all wind turbines and approximately 400 m south of the light mast, as may be seen in Figure 1. More details on the site, instrumentation and observations may also be found in Peña Diaz et al. (2016).

We consider measurements only from the western sector, with 10-minute mean wind direction between 225º and 315º. This range of wind directions is chosen for two reasons: (i) to avoid measurements from the wakes of the wind turbines and flow distortion from the mast; (ii) data from this sector corresponds to coastal and offshore conditions.

## 2.1 Selection criteria of extreme events

For the selection of the extreme variance events the 10-minute standard deviation of the wind speed measurements is compared to the extreme turbulence model in the IEC 61400-1 standard (IEC, 2005), where the horizontal turbulence standard deviation is given by

$$\sigma_1 = c \cdot I_{\text{ref}} \left[ 0.072 \left( \frac{V_{\text{ave}}}{c} + 3 \right) \left( \frac{V_{\text{hub}}}{c} - 4 \right) + 10 \right]. \tag{1}$$

Here $c$ is a constant of 2 m/s, $I_{\text{ref}}$ is the reference turbulence intensity at 15 m/s, $V_{\text{ave}}$ is the annual average wind speed at hub height, and $V_{\text{hub}}$ is the 10-minute mean wind speed at hub height; the variable of which $\sigma_1$ is a linear function of. For the

---

[1]The light mast has aircraft warning lights on the top.

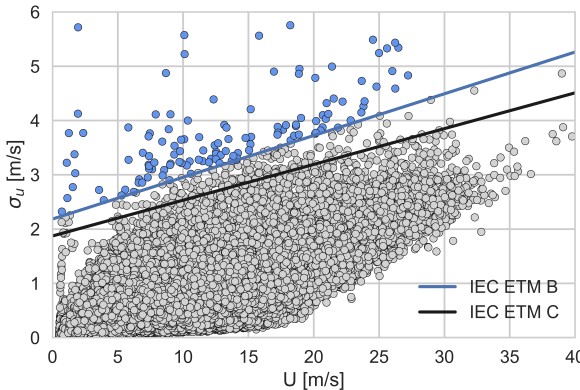

**Figure 2.** The dots correspond to 10-minute standard deviation of the wind speed as a function of $U$ at 100 m height over a 10-year period. The black and blue curves show the IEC extreme turbulence model, class C and class B respectively. The selected events (blue dots) are $\sigma_u$ values exceeding the extreme turbulence model class B.

'offshore' westerly directions considered at Høvsøre the long-term (10-year) mean of 10-minute average wind speeds at 100 m height is $U$=10.4 m/s, which corresponds well to class I turbines within the IEC 61400-1 framework with $V_{\mathrm{ave}}$=10 m/s.

The IEC standard has three turbulence categories: A, B and C, with A being the highest reference turbulence intensity, and C the lowest. The corresponding reference TI for each class may be seen in Table 1. At Høvsøre, the (decade-long) average

5   TI corresponding to the IEC reference wind speed, i.e. 10-minute mean wind speeds of $U = 15 \pm 0.5$ m/s, is below 0.12. This indicates that the reference turbulence class C and $I_{\mathrm{ref}}$ of 0.12 will equal or exceed in severity the actual conditions at the site. However, for the selection of events to analyze, a criterion corresponding to the IEC ETM model with turbulence class B is used. This is done in order to limit the selection to a representative subset of the most extreme events, while also limiting computational demands. The selected events can be seen in Figure 2 as blue dots that fall above the blue curve, i.e. these are

10   events that have a high horizontal wind speed variance. The events are selected from measurements at 100 m height.

| Turbulence class | $I_{\mathrm{ref}}$ |
| --- | --- |
| A | 0.16 |
| B | 0.14 |
| C | 0.12 |

**Table 1.** The IEC turbulence classes and associated turbulence intensities.

Figure 3 shows the horizontal wind speed at 100 m from the light mast and meteorological mast during six of the selected events. The events typically include a sudden rise in wind speed, which gives the main contribution to the high variance. Notice the sudden wind speed increase occurs approximately simultaneously at the two masts although they are ~400 m apart (for

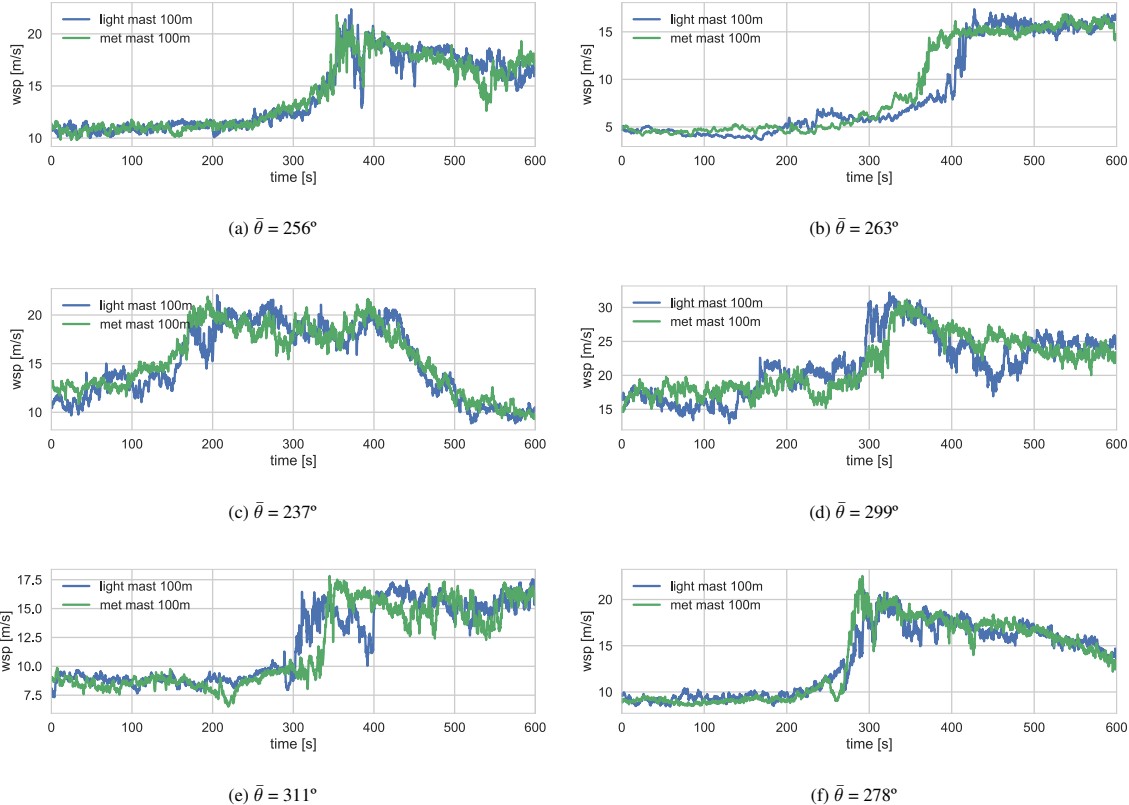

**Figure 3.** Comparison of horizontal wind speed measurements at the meteorological mast (green curve) and the light mast (blue curve). The measurement height is $100\,\text{m}$ at both masts, which are separated by $\approx 400\,\text{m}$. The 10-minute averaged wind direction $\bar{\theta}$ is from the light mast.

mean wind direction roughly perpendicular to the line connecting the masts), indicating that the events are due to large coherent structures—rather than extreme stationary turbulence.

## 3 Data processing

The data set used for the data analysis and simulation is the $10\,\text{Hz}$ measurements from cup anemometers and wind vanes on the light mast in Høvsøre.

### 3.1 Estimation of 50-year joint extremes of turbulence and wind speed: IFORM analysis

The measurements shown earlier in Figures 2 and 3 are raw (not processed or filtered), though it is common procedure to detrend data before estimating turbulence or associated return periods for a given turbulence level. Not all the extreme variance events are expected to be influenced by linear detrending, nor is such detrending necessarily appropriate for non-turbulent

events; note e.g. the event shown in Figure 3c. Therefore we want to compare the 50-year return period of turbulence with the data, detrended in two different ways: linear detrending and high-pass filtering. Detrending is performed by making a linear least-squares fit to the raw 10-minute wind speed time series, with the linear component subsequently subtracted from the raw data.

The high-pass filtering is performed with a second-order Butterworth filter (Butterworth, 1930), where the magnitude of the frequency response function (the gain) is given by

$$G(f) = \frac{1}{\sqrt{1 + (f_c/f)^4}} \tag{2}$$

where $f_c$ is the 'cut-off' frequency. We perform the filtering using a cut-off frequency of 0.0017 Hz (1/600 Hz) and also with a higher cut-off frequency of 0.0033 Hz (1/300 Hz). The higher cut-off frequency chosen for the high-pass filtering corresponds to fluctuations with periods of 300 s (half of the period of the measurements). This choice of cut-off frequency ensures removal of trends in the range 2.5–10 minutes (low-frequency transients), and is considered conservative enough to still include fluctuations associated with turbulent eddies.[2]

Here we use the inverse first-order reliability method (IFORM) to estimate the 50-year return period contour corresponding to the joint description of turbulence ($\sigma_u$) and 10-minute mean wind speed ($U$). This method was developed by Winterstein et al. (1993) and provides a practical way to evaluate joint extreme environmental conditions at a site. The IFORM method is widely used in wind energy to predict extreme environmental conditions or long-term loading on wind turbines, for ultimate strength analysis. More information on this method may be found in e.g. Fitzwater et al. (2003); Saranyasoontorn and Manuel (2006); Moon et al. (2014).

The first step in the IFORM analysis is to find the joint probability distribution $f(U, \sigma_u)$. According to the IEC standard the 10-minute mean wind speed is assumed to follow a Weibull distribution [3], and the 'strength' (standard deviation) of turbulent stream-wise velocity component fluctuations ($\sigma_u$) is assumed to be log-normally distributed conditional on wind speed. In the standard, the mean of $\sigma_u$ is expressed as a function of $U$,

$$\mu_{\sigma_u} = I_{\text{ref}}(0.75U + 3.8\,\text{m/s}), \tag{3}$$

and the standard deviation of $\sigma_u$ is defined as

$$\sigma_{\sigma_u} = 1.4 I_{\text{ref}}. \tag{4}$$

In Figure 4, $\mu_{\sigma_u}$ and $\sigma_{\sigma_u}$ are shown as functions of 10-minute mean wind speed, from Høvsøre unprocessed measurements at 100 m (grey dots) and the expressions from the IEC standard (blue lines) with $I_{\text{ref}} = 0.12$. The green lines show a third- and a second order polynomial fit to the binned measurements of $\mu_{\sigma_u}$ and $\sigma_{\sigma_u}$ respectively (bins of 1 m/s). The IEC expression

---

[2] Fluctuations with a period of 300 s at 4 m/s–25 m/s (the operational wind speed range of a typical wind turbine) correspond to length scales of 1200 m–7500 m. Length scales in this range are significantly larger than turbulent length scales that have been estimated at the Høvsøre site (e.g. Sathe et al., 2013; Dimitrov et al., 2017; Kelly, 2018)

[3]Here we use a 3-parameter Weibull distribution. This is done because after filtering out measurements with errors and missing periods, the lowest mean wind speed is 2.2 m/s. One could also use a weighted 2-parameter Weibull distribution fit with increased weights in the tail to obtain the same result.

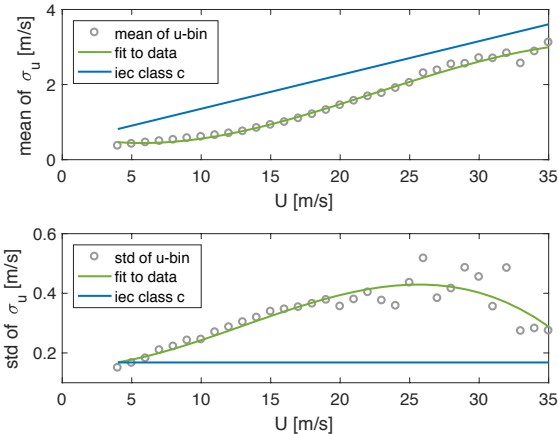

**Figure 4.** The mean and standard deviation of $\sigma_u$ as function of wind speed at 100 m, for raw data (not de-trended or filtered). The blue curves show the IEC expressions, the grey dots show the measured values and the green curves show a polynomial fit to the measurements.

for $\mu_{\sigma_u}$ is higher than that from the measurements, but has a similar slope for mean wind speeds above 15 m/s. The difference is larger between the data and IEC expression for $\sigma_{\sigma_u}$, where the assumption of no mean wind speed dependency does not fit well to the data.

The next step in the IFORM analysis is to obtain a utility "reliability index" $\beta$ which translates the desired return period $T_r$
(here 50-years) into a normalized measure corresponding to number of standard deviations of a standard Gaussian distribution:

$$\beta = \Phi^{-1}\left(1 - \frac{T_t}{T_r}\right) = \Phi^{-1}\left(1 - \frac{1}{5n_m}\right) \tag{5}$$

Here $\Phi^{-1}$ is the inverse Gaussian cumulative distribution function (cdf), $T_t$ is the duration of a turbulence measurement (here 10 minutes) and $n_m$ is the number of 10-minute measurements corresponding to a 10-year period (which equals the time span
of the data). Thus the reliability index equals the radius of a circular contour in standard Gaussian space, so that

$$\beta = \sqrt{u_1^2 + u_2^2}, \tag{6}$$

where the standard normal variables $u_1$ and $u_2$ are derived from physical variables using an iso-probabilistic transformation, which takes correlations into account. We invoke the Rosenblatt transformation (Rosenblatt, 1952), which relies on the fact that a multivariate distribution may be expressed as a product of conditional distributions: $F(x_1, x_2) = F(x_1)F(x_2|x_1)$. In this
analysis, only two variables are considered, and the transformation may be performed in the following way:

$$U = F_U^{-1}\Big(\Phi(u_1)\Big) \quad , \quad \sigma_u = F_{\sigma_u|U}^{-1}\Big(\Phi(u_2)\Big) \tag{7}$$

where $F_U$ is the three-parameter Weibull cdf and $F_{\sigma_u|U}$ is the conditional log-normal cdf.

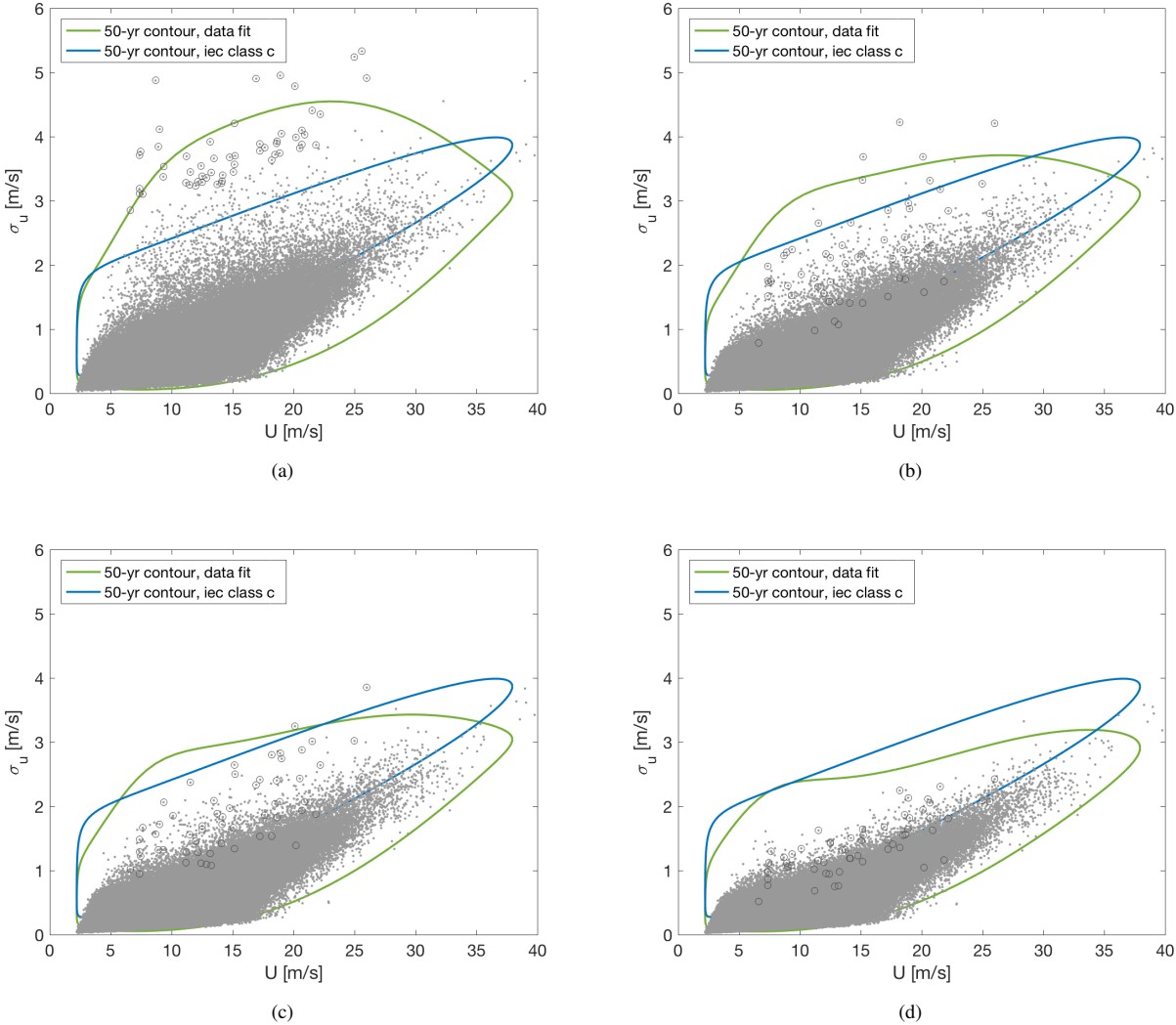

**Figure 5.** The 50-year return period contours based on the measurements (green curves) and the IEC expressions (blue curves). The grey dots show the measurements. a) Raw measurements. b) Linearly detrended measurements. c) High-pass filtered measurements with a cut-off frequency: 1/600 Hz. d) High-pass filtered measurements with a cut-off frequency: 1/300 Hz. The dark grey circles indicate the extreme variance events.

Figure 5 shows the joint distribution of mean wind speed and turbulence[4], with contours corresponding to the 50-year return period. The contours are calculated based on the measurements (green curves), and the IEC expressions (blue curves) of $\mu_{\sigma_u}$ and $\sigma_{\sigma_u}$ respectively. The parameters of the marginal distribution of the 10-minute mean wind speed data were found with maximum likelihood estimation of the three-parameter Weibull distribution (scale parameter: 9.75 m/s, shape parameter: 2.02,

---

[4]Note that some measurement points have been removed due to measurement errors, therefore the points are fewer than in Figure 2, which includes 10-minute statistics from the whole measurement period.

location parameter: 2.20). The parameters for the conditional log-normal distribution were estimated with the first and second moments, conditional on mean wind speed: $\mu_{\sigma_u}$ and $\sigma_{\sigma_u}$, both with the IEC expressions in Eqs. 3 and 4 and the third- and second-order polynomial fit to the binned data. It is seen when comparing Figures 5a to 5d that the variance of $\sigma_u$ is significantly reduced by the high-pass filtering. The 50-year return period contour estimated with the linearly detrended data (Figure 5b) exceeds the one estimated with IEC turbulence class C in the whole operational wind speed range. This is because the linear detrending does not affect events like the one seen in Figure 3c, and these events influence the estimate of the contour. Figure 5c shows the high-pass filtered measurements with a cut-off frequency of 1/600 Hz, and here it is seen how the estimated 50-year return period contour exceeds the IEC turbulence class C contour for wind speeds between 6 m/s and 22 m/s. In Figure 5d, it is seen how the high-pass filtering with cut-off frequency of 1/300 Hz reduces the variance estimates to the extent that the 50-year contour obtained in this way gives turbulence levels lower than ETM IEC class C. These observed changes in turbulence levels indicate that the extreme variance events are not necessarily associated with linear trends. Some events are associated with wind speed fluctuations in a frequency range that may have a substantial impact on wind turbine loads. Therefore we investigate this impact, with constrained turbulence simulations incorporating the raw measurements that have not been detrended in any way.

## 3.2 Time series for simulation

The peak and the corresponding location of each event is identified in the following way: A moving average is subtracted from the wind speed signal and the maximum value of the differences identified:

$$u_{\text{peak}} = \max(u - \bar{u}_{60s}) \tag{8}$$

where $u$ is the horizontal wind speed signal and $\bar{u}_{60s}$ is the moving average over 60 s. The peaks are not necessarily the highest value of the signal, but rather the highest value within a sharp wind speed increase.

Applying the selection criteria described in section 2.1 results in 99 identified events. Of these, 30 events are discarded as they include periods of missing measurements. A lower threshold of 4 m/s is put on $u_{\text{peak}}$ to exclude events mostly consisting of a linear trend or relatively insignificant peaks. Finally, events where the corresponding directional data fluctuated below 180º are discarded, i.e. temporary directional data from South, to exclude measurements from the wake of the nearby wind turbine. A remaining 44 events are chosen for load simulations. The measured time series including the extreme events are used to generate constrained turbulence simulations (explained in more detail in Section 4.4) of 600 s duration. The time series period is selected such that the sharp wind increase, or ramp, occurs approximately in the middle of the time series, i.e., approximately 300 s before and after the peak.

## 4 Load simulation environment

### 4.1 HAWC2 and the DTU 10 MW

Wind turbine response in the time domain is calculated with HAWC2 (Horizontal Axis Wind turbine simulation Code 2nd generation, Larsen and Hansen, 2015). HAWC2 is based on a multibody formulation for the structural part, where each body

consists of Timoshenko beam elements. All the main components of a wind turbine are represented by these independent bodies and connected with different kinds of algebraic constraints. The aerodynamic forces are accounted for with blade element momentum theory (see e.g. Hansen, 2013) with additional correction models: a tip correction model, a skewed inflow correction, and a dynamic inflow correction. HAWC2 additionally includes models that account for dynamic stall, wind shear effects on induction, tower-induced drag, and tower shadow.

All the load simulations are performed using the DTU 10 MW reference wind turbine (RWT), which is a virtual wind turbine model based on state-of-the-art wind turbine design methodology. The main characteristics of the RWT may be seen in Table 2 and a more detailed description may be found in Bak et al. (2013). The controller used for the RWT is the Basic DTU Wind Energy controller (Hansen and Henriksen, 2013).

| DTU 10 MW RWT | |
| --- | --- |
| Rotor diameter | 178.3 m |
| Cut-in wind speed | 4 m/s |
| Rated wind speed | 11.4 m/s |
| Cut-out wind speed | 25 m/s |
| Cut-in rotor speed | 6 rpm |
| Rated rotor speed | 9.6 rpm |
| Hub height | 119 m |

**Table 2.** The main characteristics of the reference wind turbine.

## 4.2 Turbulence simulations in HAWC2

The Mann spectral turbulence model (Mann, 1994, 1998) is fully integrated into HAWC2, where a turbulence 'box' may be generated for every wind turbine response simulation. The turbulence box is a three dimensional grid that contains a wind velocity vector at each grid point. The turbulence boxes in this study all have $8192 \times 32 \times 32$ grid points, in the $x$-, $y$-, and $z$-directions, respectively. The $y$-$z$ plane is parallel with the rotor, and the distance between the grid points is typically defined so that the domain extent in the $y$- and $z$-directions becomes a few percent larger than the rotor diameter. The length of the $x$-axis ($L_x$) is proportional to the mean wind speed at hub height, $L_x = U \cdot T$, where $T$ is the simulation time. The turbulence box is transported with the average wind speed at hub height through the wind turbine rotor.

The Mann model is based on an isotropic von Kármán turbulence spectral tensor, which is distorted by vertical shear caused by surface friction. Assumptions of constant shear and neutral atmospheric conditions in the rapid-distortion limit are used to linearize the Navier-Stokes equations, which may then be solved as simple linear differential equations. The solution results in a spectral tensor that may be used in a Fourier simulation, to generate a random field with anisotropic turbulent flow. The Mann model contains three parameters:

- $\Gamma$ is an anisotropy parameter, that when positive, $\sigma_u^2 > \sigma_v^2 > \sigma_w^2$, which are the variances of the $u$-, $v$- and $w$-components of the wind speed, respectively. When $\Gamma = 0$, the generated turbulence is isotropic, $\sigma_u^2 = \sigma_v^2 = \sigma_w^2$.

- $\alpha\varepsilon^{2/3}$ is the product of the Kolmogorov spectral constant and the rate of turbulent kinetic energy dissipation to the power of 2/3. The Fourier amplitudes from the spectral tensor model are proportional to $\alpha\varepsilon^{2/3}$, hence increasing $\alpha\varepsilon^{2/3}$ gives a proportional increase in the simulated turbulent variances, but no change in the shape of the spectrum.

- $L$ is the length scale which is representative of the eddy size that contains the most energy.

The IEC-recommended values of the parameters are: $\Gamma = 3.9$, $L =$29.4 m (for hub heights above 60 m), and that $\alpha\varepsilon^{2/3}$ is set to a positive value, to be scaled with $\sigma_u^2$. It has been shown in numerous studies that these parameters can change significantly, e.g. with turbulence level (Dimitrov et al., 2017; Kelly, 2018), atmospheric stability (Sathe et al., 2013; Chougule et al., 2017) and site conditions (Kelly, 2018; Chougule et al., 2015). As we do not want to investigate the effect of changing these parameters, all turbulence realizations are chosen to have the same parameters. In the present study, the anisotropy parameter is chosen according to the IEC standard, $\Gamma = 3.9$. The turbulence length scale is chosen differently, because the DTU 10 MW RWT is a relatively large wind turbine, and the turbulence length scale is expected to be of the same order of magnitude as the hub height (Kristensen and Frandsen, 1982). Here the length scale is estimated via

$$L = \frac{\sigma_u}{dU/dz} \tag{9}$$

as derived by Kelly (2018). The final 200 s of simulation data, i.e. after the wind-speed ramps, are used to estimate the length scale of turbulence and thus exclude the large coherent structure. Here $\sigma_u$ from 100 m height is used, along with $dU/dz$ estimated between $z = 160$ m and $z = 60$ m. Using (9) the length scale is found on average to be $\langle L \rangle \approx 120$ m over all events analyzed. The value chosen is therefore $L = 120$ m.

## 4.3 Design load case 1.3

The DLC is simulated based on the setup described in Hansen et al. (2015), where mean wind speeds at hub height of 4–26 m/s in steps (bins) of 2 m/s are used, and each simulation has a duration of 600 s[5]. The Mann turbulence model is used to generate Gaussian turbulence boxes, with six different synthesized turbulence seeds per mean wind speed. The simulation time of the turbulence boxes is defined to be 700 s, where the first 100 s are used for initialization of the wind turbine response simulation, and are disregarded for the load analysis.

## 4.4 Constrained turbulence simulations

The aim here is to generate turbulence simulations resembling the measured wind field of the extreme variance events. This is done by constraining the synthesized turbulence fields. The constraining procedure involves modifying the time series to represent the most likely realization of a random Gaussian field which would satisfy the constraints, using an algorithm described

---

[5]In contrast with Hansen et al. (2015), here the simulations are performed without yaw misalignment.

in Hoffman and Ribak (1991) and demonstrated with applications to wind energy in Nielsen et al. (2004) and Dimitrov and Natarajan (2017). For the constraining procedure we define three different random Gaussian fields as a function of location, $\mathbf{r} = \{x, y, z\}$:

1. the constrained field, $f(\mathbf{r})$, which is the generated field of the procedure, modified to resemble the measurements;

2. the source field, $\tilde{f}(\mathbf{r})$, which here is a random realization of the Mann turbulence model;

3. the residual field, which is the difference between the constrained field and the source field, $g(\mathbf{r}) = f(\mathbf{r}) - \tilde{f}(\mathbf{r})$.

The constraints are a set of $M$ values at given locations, $C = \{c_1(\mathbf{r}_1), c_2(\mathbf{r}_2), ..., c_M(\mathbf{r}_M)\}$, which the constrained field is subject to, i.e. $f(\mathbf{r}_i) = c_i$. At the constraint points, the residual field is given by $g(\mathbf{r}_i) = c_i - \tilde{f}(\mathbf{r}_i)$, and for all other locations the values are conditional on the constraints in $C$. The conditional probability distribution of the residual field is denoted by the multivariate Gaussian distribution function

$$\phi\big(g(\mathbf{r})|C\big) = \frac{\phi(g(\mathbf{r}), C)}{\phi(C)}. \tag{10}$$

The conditional probability function of the field may be described as a shifted Gaussian around the conditional ensemble average $\langle g(\mathbf{r})|C \rangle$, where

$$\langle g(\mathbf{r})|C \rangle = R_i(\mathbf{r}) R_{ij}^{-1}(C - \tilde{f}[\mathbf{r} = \mathbf{r}(c_i)]) \tag{11}$$

where $\langle ... \rangle$ is the ensemble average, $R_i(\mathbf{r}) = \langle f(\mathbf{r}) C_i \rangle$ are the cross-correlations between the field and constraints, $R_{ij} = \langle C_i C_j \rangle$ are the correlations between the constraints, and $\tilde{f}[\mathbf{r} = \mathbf{r}(c_i)]$ are the values of the source field at the constraints' locations.

A realization of the constrained field is generated by adding the conditional ensemble mean of the residual field to the source field

$$f(\mathbf{r}) = \tilde{f}(\mathbf{r}) + \langle g(\mathbf{r})|C \rangle \tag{12}$$

Here the constraints consist of the $u$- and $v$-components of the wind velocity measurements from the light mast. The constraints are applied at three different heights : 79 m, 119 m (hub height) and 179 m, i.e. shifted up 19 m so the measurements at 100 m represent hub height wind speed. The constraints are also applied at three different widths (along the y-axis): 89.6 m (the middle of the turbulence box) $\pm70$ m. This is done to ensure the coherent structure of the observed flow in the simulations. Every third measurement is applied at each width along the y-axis, giving applied constraints at each y-location with a 3.33 Hz frequency. This is done to reduce the number of applied constraints and thereby the computational time of the simulations.

In Figure 6 two turbulence boxes with different random seeds are seen. The u-component of the turbulent field is shown with a color scale on slices along the time axis. The upper plots show the unconstrained turbulence boxes, and the lower plots show the same turbulence boxes with constraints corresponding to measurements from two different extreme variance events.

Figure 7 shows two examples of the u-velocity time series at hub height with and without applied constraints, for the same turbulence seeds as shown in Figure 6.

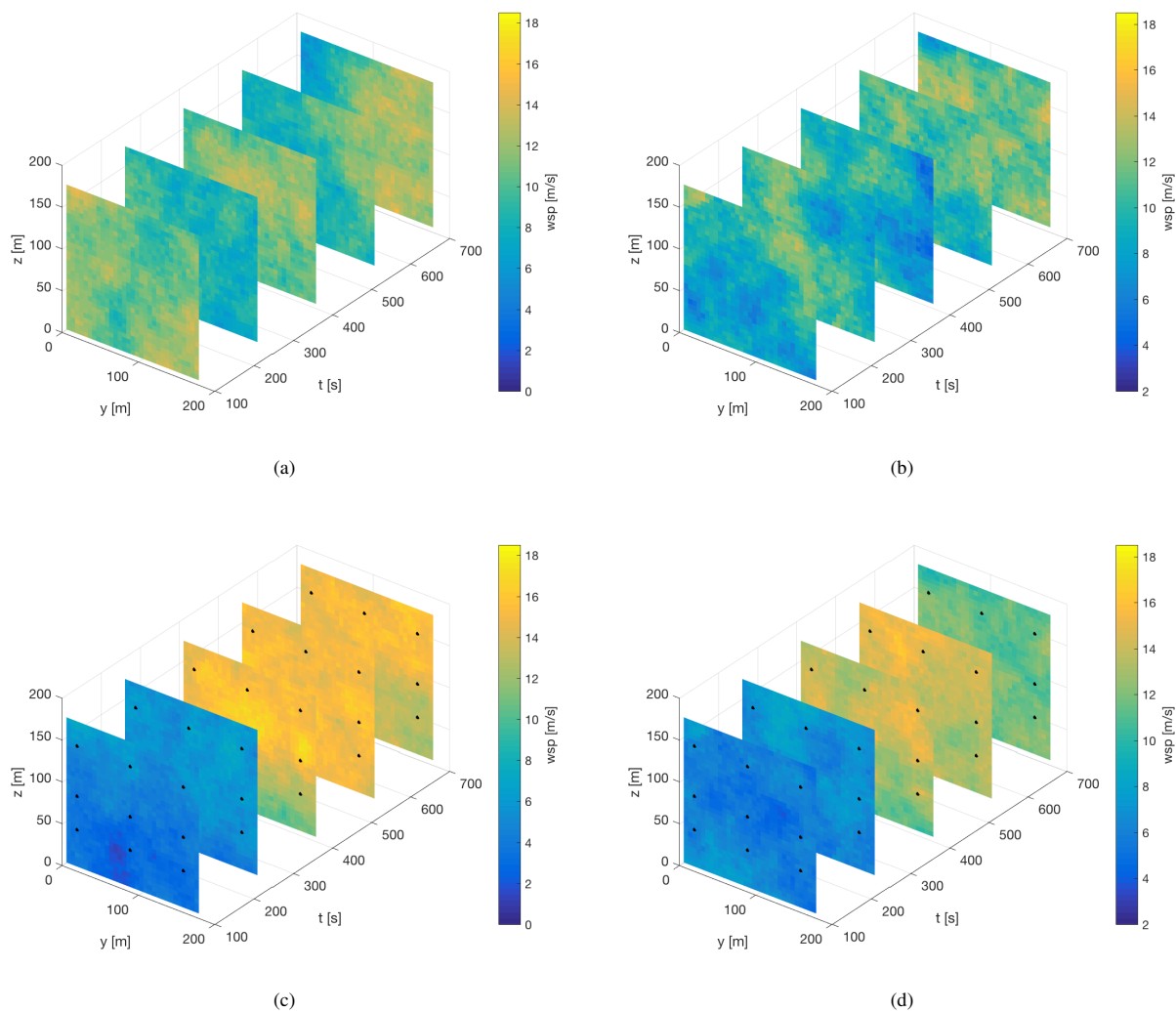

**Figure 6.** Comparison between u-velocity components from unconstrained turbulence simulations, and from turbulence simulations with velocity jumps included using constrained simulation. a) Seed 1003 without constraints. b) Seed 1005 without constraints. c) Seed 1003 with constraints. d) Seed 1005 with constraints. Constraint locations are shown with black dots.

For the purpose of load simulations, six different constrained turbulence seeds are generated from each extreme variance event time series. Although applying the constraints makes the turbulence boxes similar in general, there are differences in the parts of the boxes which are far from the constraint locations. As a result, there will be a seed-to-seed variation in loads simulated with constrained turbulence boxes, albeit much smaller than what is seen in the unconstrained case.

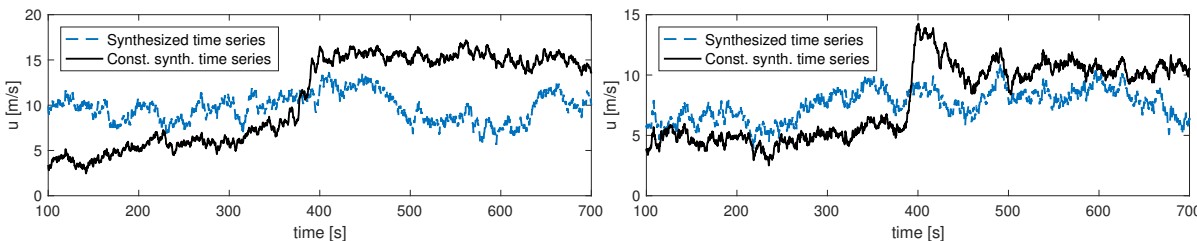

**Figure 7.** Comparison of unconstrained and constrained stream-wise ($u$-) velocity component in the middle of the turbulence box, y=89.6 m, z=119 m. *Left:* Seed 1003. *Right:* Seed 1005.

# 5 Load simulation results

In this section we compare the design load levels of the two simulation sets: DLC 1.3 and the constrained simulations with the extreme variance. DLC 1.3 consists of 72 simulations (6 seeds per 12 wind speed bins) and the constrained simulations consist of 264 simulations (6 seeds per 44 extreme variance event).

## 5.1 Extreme loads

In Figure 8 the standard deviation of the simulated hub height u-component wind speed is shown as function of the mean hub height u-component wind speed. Each dot shows the standard deviation averaged over six turbulence seeds. As the variance is scaled to match the target both for DLC 1.3 and the constrained simulations, the scatter of the mean standard deviation over the six different seeds is small. The standard error of the mean standard deviation is in the range of 0.008 - 0.013 m/s, and the standard error of the mean hub-height u-component wind speed is equal to, or less than 0.015 m/s. The standard deviation from the constrained turbulence simulations (blue dots) is higher than that of DLC 1.3 with one exception. For this case, some variance was lost as a consequence of changing the time interval selection to span $\pm 300$ s around the wind speed peak, and data with a negative trend was cut off.

In Figure 9 the characteristic extreme loads from DLC 1.3 and the constrained simulations are compared. The maximum/minimum load values of each 10-minute HAWC2 simulation are binned according to wind speed with a bin width of 2 m/s and then averaged. For the comparison we omit the wind speed bin at 26 m/s, as there are no observed events within that wind speed bin. The error bars show the standard deviation of the extreme loads of each wind speed bin. Both maxima and minima are shown for the tower-top moments, but for all other load components only the maximum moments are shown. It should be noted that the in-plane blade root flap moment maxima are negative, due to the orientation of the blade coordinate system of the wind turbine model in HAWC2.

The two top panels show the extremes of the tower top tilt and yaw moments, respectively. In the whole wind speed range the mean extreme moments for DLC 1.3 are between 6400 - 21000 kNm larger than for the constrained simulations.

The left middle panel shows the mean extreme tower base fore-aft moments. The overall highest mean extreme moment is from the DLC 1.3 simulation set, however for the constrained turbulence simulations the loads are higher for wind speed bins

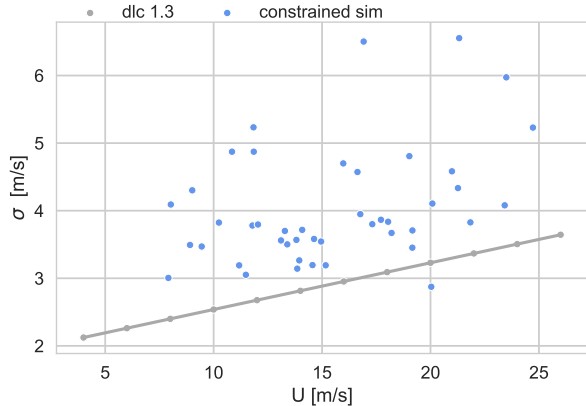

**Figure 8.** The mean standard deviation of the u-component of the simulated wind speed at hub height as function of mean wind speed at hub height. DLC 1.3 (grey dots) and constrained simulations with extreme variance events (blue dots).

at 8 m/s and between 14-20 m/s. The largest difference is seen for wind speed bin 16 m/s where the mean extreme moment from the constrained simulation is 50200 kNm larger than from the DLC 1.3.

The right middle panel shows the mean extreme tower base side-side moments. In the whole wind speed range the mean extreme moments for the DLC 1.3 are between 6000 - 22500 kNm larger than for the constrained simulations.

5    The two bottom panels show the blade root- flap and and edge moments respectively. In the whole wind speed range the mean extreme moments for the DLC 1.3 are between 800 - 6200 kNm larger than for the constrained simulations, with the exception of wind speed bin 16 m/s, where the mean extreme moments from the constrained simulations are respectively 3000 kNM and 400 kNm higher than the DLC 1.3.

The extreme tower top tilt-, yaw- and tower base side-side moments show a general increase with wind speed. The extreme

10    blade root flap- and tower base fore-aft moments peak around rated wind speed. For the extreme blade root edge moment it is seen that the loads peak around rated wind speed for both simulation sets, but the main difference is that after 16 m/s the DLC 1.3 loads and the scatter increases with wind speed.

| Mean extreme moment | DLC 1.3 [kNm] | Constrained sim [kNm] | Ratio (Const./DLC) |
|---|---|---|---|
| Tower top tilt | $3.08 \cdot 10^4$ | $1.83 \cdot 10^4$ | 0.60 |
| Tower top yaw | $-3.07 \cdot 10^4$ | $-1.21 \cdot 10^4$ | 0.40 |
| Tower base fore-aft | $2.20 \cdot 10^5$ | $2.14 \cdot 10^5$ | 0.97 |
| Tower base side-side | $6.38 \cdot 10^4$ | $4.12 \cdot 10^4$ | 0.65 |
| Blade root flap | $-3.91 \cdot 10^4$ | $-3.51 \cdot 10^4$ | 0.90 |
| Blade root edge | $1.55 \cdot 10^4$ | $1.29 \cdot 10^4$ | 0.83 |

**Table 3.** The highest mean extreme moments for different load components

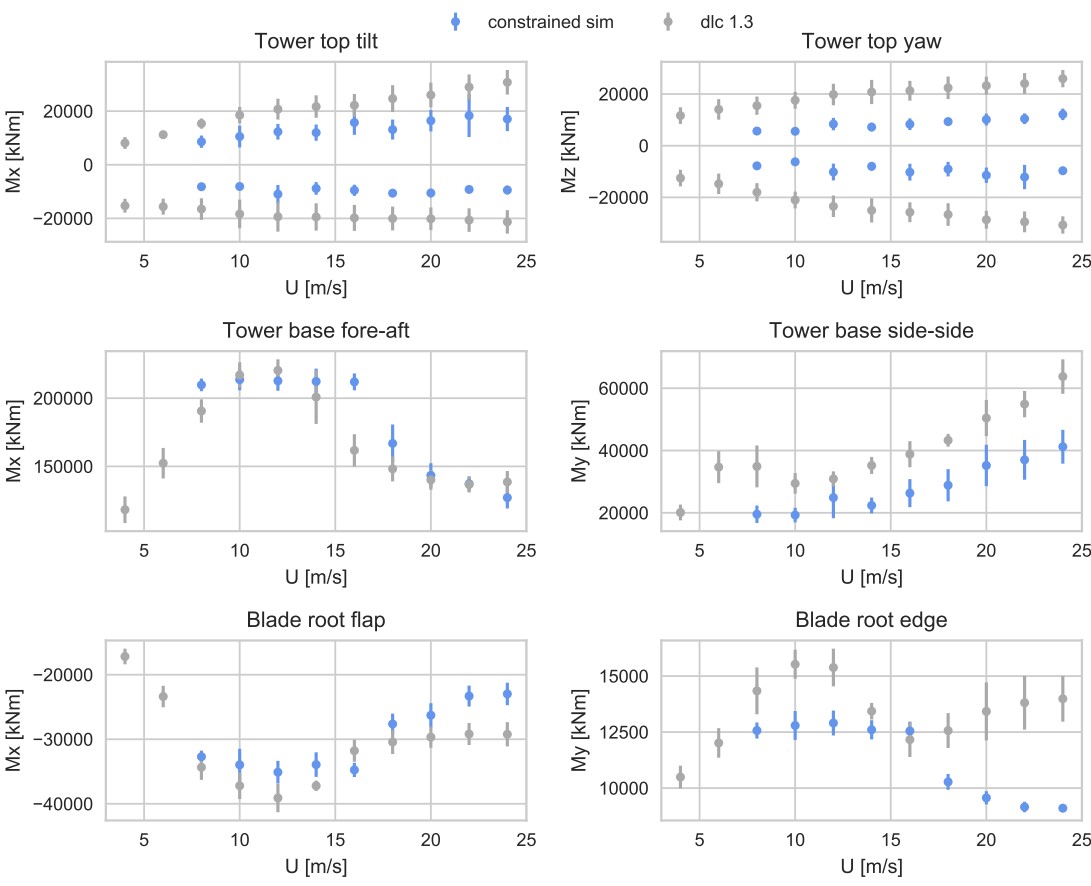

**Figure 9.** The mean extreme moments from IEC DLC 1.3 (grey dots). The mean extreme loads from the constrained simulations (blue dots).

Table 3 lists the overall characteristic loads from each simulation set (the extremes seen in Figure 9), together with their ratio. The difference between the overall extremes from the two simulation sets is largest for the tower-top yaw moment, where the extremes are lower from the constrained simulations. The overall extremes are of similar magnitude for the tower base fore-aft moment and the blade root flap-wise moment.

## 5.2 Time series of turbine loads

In the following, examples of 10-minute time series from DLC 1.3 and constrained simulation sets are shown side by side, for comparison and demonstration of the differences in the wind turbine response to different types of wind regime. A comparison is made for the tower-base fore-aft moment, where the characteristic extreme loads from the different simulation sets are of similar magnitude. We also consider and compare the tower top tilt- and yaw-moments, which give the largest differences between the two simulation sets.

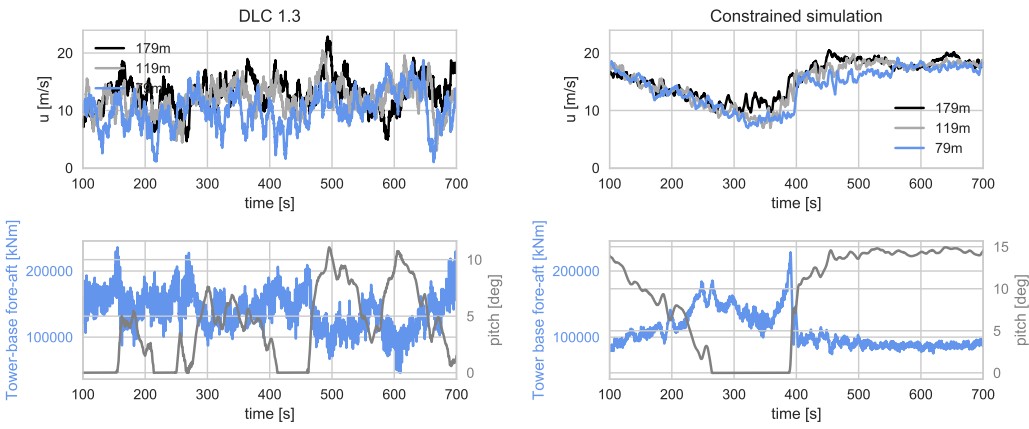

**Figure 10.** Comparison of a DLC 1.3 time series (left panels) and a constrained simulation time series of an extreme event (right panels). *Top panels*: $u$-component wind speed. *Bottom panels*: Tower-base fore-aft moment (blue) and pitch angle (grey).

First, we compare two time series giving some of the highest extreme tower base fore-aft moments from each simulation set. For DLC 1.3 in Figure 10 the mean $u$-component hub-height wind speed is $U = 12.0$ m/s, with standard deviation of $\sigma_u = 2.7$ m/s and the peak tower base fore-aft moment is 236000 kNm. For the constrained simulation, $U = 14.9$ m/s and $\sigma_u = 3.5$ m/s. The peak tower base fore-aft moment is 228000 kNm. The peak tower base fore-aft moments are of similar

magnitude in the simulations, and in both cases this occurs when the pitch angle is zero degrees—right before the wind turbine blades begin to pitch. Also, at the time when the wind speed at hub height reaches rated wind speed, the wind speed at 179 m is above rated wind speed, leading to higher loading on the upper half of the rotor. From the turbulence simulations, the most noticeable difference in the wind turbine response is that in the constrained turbulence simulation the time of the peak tower base fore-aft moment is very distinguishable at 390 s. While for the stationary turbulence the peak response occurs around

150 s, but numerous times it reaches above 200000 kNm during the simulation. Note that the axes in the top panels are the same, as are the axes in the bottom panels. It is seen that although the standard deviation of the wind speed is lower in the stationary turbulence simulation, the wind speed extremes are greater, with instantaneous wind speed reaching below 2 m/s and above 22 m/s.

In Figure 11 we compare some of the most extreme tower top moments from the two simulation sets. The stationary turbu-

lence simulation in Figure 11, has $U = 22$ m/s, $\sigma_u = 3.4$ m/s, with a peak tower top tilt moment of 36601 kNm and a peak tower top yaw moment of $-28900$ kNm; in contrast the constrained turbulence simulation has $U = 21.3$ m/s, $\sigma_u = 6.6$ m/s, with a peak tower top tilt moment of 30800 kNm and a peak tower top yaw moment of $-18600$ kNm. As in the previous example, the time of peak loads is very clearly identified in the constrained turbulence simulation, and the peak value is significantly higher than the response for the remainder of the simulation. For the stationary turbulence simulation, the tower top yaw- and tilt mo-

ments often reach high values throughout the simulation. Extreme tower-top moments tend to be observed when there is high shear across the rotor. In stationary turbulent flow the variation in wind speed across the rotor arises as turbulent eddies sweep

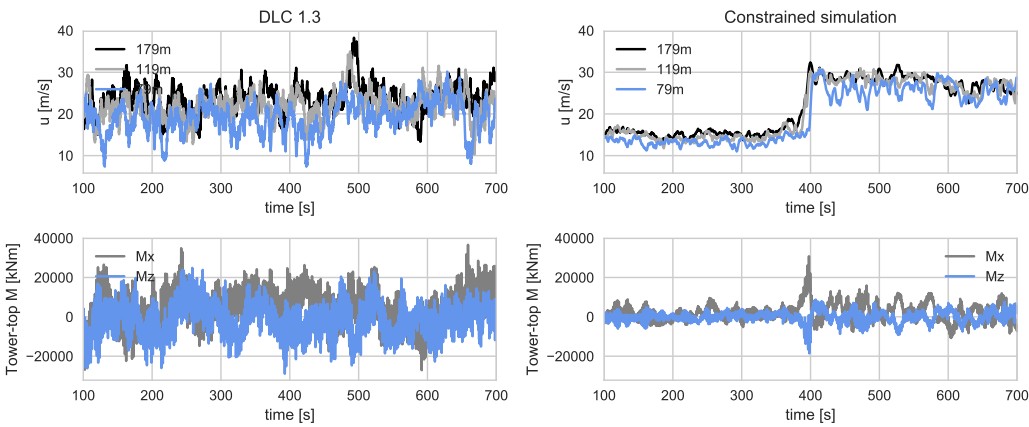

**Figure 11.** Comparison of a DLC 1.3 time series (left panels) and a constrained simulation time series of an extreme event (right panels). *Top panels*: $u$-component wind speed. *Bottom panels*: Tower-top tilt (grey) and yaw (blue) moments.

by, hitting only part of the rotor, leading to high wind shear. The extreme tower top loads from the constrained simulations are in connection with high vertical wind shear arising during the wind speed increase (ramp event).

## 6 Discussion

In the load time series comparison, the general differences in the wind turbine response of the two simulation sets are visualized;
for the constrained simulations the peak loads are distinguishable and occur because of the velocity increase associated with the ramp-like event. The discrepancies between the two simulation sets for the extreme tower top loads indicate that the short-term wind field variability across the rotor is generally higher in the stationary turbulence simulation than for the constrained simulations. As shown in the time series comparison of Figure 11, the short-term vertical wind shear can be high in connection with the extreme events, yet the tower top tilt moment does not exceed that prescribed via DLC 1.3. When non-uniformity in
the stationary turbulence fields occurs around rated wind speed, it can also lead to high extreme tower base fore-aft moments that are connected to high thrust on the rotor. The extreme tower base fore-aft moments from the constrained simulations are highest for mean wind speed bins between 8 m/s and 16 m/s. In this wind speed range, the wind speed is typically below rated wind speed at the beginning of the simulation and later increases beyond rated wind speed. When the wind speed starts to rise, it does so coherently across the rotor plane, resulting in high thrust and tower base fore-aft moments, before the wind turbine
controller starts to pitch the blades. The tower base fore-aft moments for the extreme turbulence case (IEC DLC 1.3) were expected to be lower than those of the extreme variance events; however, this was generally true only (on average) for certain wind speed bins. The overall characteristic tower base fore-aft moment of DLC 1.3 is 3% higher than for the extreme events.

The load simulation results show that the extreme turbulence case DLC1.3 indeed covers the load envelope caused by extreme variance events. However, the differences seen in the time series and in the load behavior indicates that extreme

variance observations as events are entirely different from situations with stationary, homogeneous turbulence. This questions the basis for the definition of the IEC Extreme Turbulence Model (ETM) which is defined in terms of the statistics of the 10-minute standard deviation of wind speed. As most observations of the selected extreme variance events include a short term ramp event, it would perhaps be more relevant to compare these events with other extreme design load cases in the IEC standard, e.g. the extreme coherent gust with direction change, extreme wind shear or the extreme operating gust. Since these are the absolute highest variance events observed at Høvsøre during a ten year period, they would also appear in the site-specific definition of the ETM model. Therefore, it may be necessary to exclude or re-assign such events to the relevant load case type. The design and cost of a wind turbine may depend on how this consideration is done.

In the current study we generate Gaussian turbulence fields only, though it is known that atmospheric turbulence can exhibit some non-Gaussian character (e.g., Peinke et al., 2004; Wilczek and Friedrich, 2009; Morales et al., 2012). But the extent to which the non-Gaussian aspect impacts the response dynamics of wind turbines is the subject of ongoing debate. Studies have shown non-Gaussian wind fields to impact the loads on and output of wind energy converters; e.g., the torque fluctuations of a numerical wind turbine model (Mücke et al., 2011), the power and torque of a model wind turbine in a wind tunnel experiment (Schottler et al., 2016) and the power of a full-scale 2.5 MW wind turbine (Chamorro et al., 2015). However, a recent study based on large-eddy simulations of atmospheric turbulence shows that Gaussian and non-Gaussian turbulence, as input to wind turbine load simulations, result in insignificant differences (Berg et al., 2016). The conditions under which non-Gaussianity can significantly affect turbines (loads and power) still remain to be determined in ongoing research. The main focus of the current study is non-stationary ramp events and their impact on wind turbine loads, rather than comparison of Gaussian and non-Gaussian turbulence fields upon which the ramps are superposed. We use generated Gaussian turbulent fields as they are readily available, recommended by the IEC standard and to restrict the complexity of the study. Further, the loads are dominated by the ramp events and not by the turbulence.

It was seen in the IFORM analysis in section 3.1 that the estimated 50-year return period contour of the linearly detrended data exceeded the 50-year return period contour of normal turbulence (corresponding to the ETM class C). This is consistent with the findings of Dimitrov et al. (2017), who performed similar analysis of linearly detrended measurements from Høvsøre, though from the easterly (homogeneous farmland) sector. For the high-pass filtered measurements, the turbulence level was reduced significantly as well as the estimated 50-year return period of turbulence. This is seen as the high-pass filtering effectively removes variance of low frequency fluctuations with time scales larger than 300 s, as the chosen cut-off frequency was 1/300 Hz. This finding suggests that for typical hub heights as considered ($z \approx 100$ m) at a coastal site like Høvsøre, extreme variance events are not representative of homogeneous, stationary turbulence and can be filtered out by high-pass filtering. It should be kept in mind though, that these events may be considered for extreme design load case purposes other than turbulence. In that case it is important not to use detrending of any kind on the measurements, as these extreme fluctuations will then not be identified and characterized correctly.

## 7 Conclusions

The main objective of this study is to investigate how extreme variance events influence wind turbine response and how it compares with DLC 1.3 of the IEC 61400-1 standard. The selected extreme events are measurements of the 10-minute standard deviation of horizontal wind speed that exceed the values prescribed by the ETM model and include a sudden velocity jump (ramp event, transients in the turbulent flow), which is the main cause of the high observed variance. The events were simulated with constrained turbulence simulations, where the measured time series were incorporated in turbulence boxes for load simulations in order to make a realistic representation of the events, including the short term ramps and the coherent flow in the lateral direction as was seen in the comparison of measurements between the two masts in Figure 2. The constraints force the turbulent flow of the simulations to be non-stationary and non-homogeneous.

Load calculations of the simulated extreme events were made in HAWC2 and compared to load calculations with stationary homogeneous turbulence according to DLC 1.3. To summarize, we have found that:

- The extreme variance events are large coherent structures, observed simultaneously at two different masts with a 400 m (lateral) separation.

- Most extreme variance events include a sharp wind speed increase (short-time ramp) which is the main source of the large observed variance.

- High-pass filtering with a cut-off frequency of 1/300 Hz removes most of the variance corresponding to these ramp-like events, to the extent that the estimated 50-year return period of (remaining) turbulence level is lower than that of IEC ETM class C; linear de-trending may remove some of the variance but is not necessarily adequate.

- Compared with the DLC 1.3 of the IEC standard, the extreme loads are on average lower for the extreme variance events in the coastal/offshore climate and heights considered.

- For 10-minute mean wind speeds of 8–16 m/s, the events typically begin below rated wind speed and increase beyond, leading to high thrust on the rotor; such events lead to high extreme tower-base fore-aft loads which can exceed the DLC 1.3 prescription of the IEC standard.

Future related work includes further analysis and characterization of extreme variance events. In particular, ongoing work involves extreme short-term shear associated with such events, and directional change. Load simulations of the events may be compared with other extreme DLC's from the IEC standard.

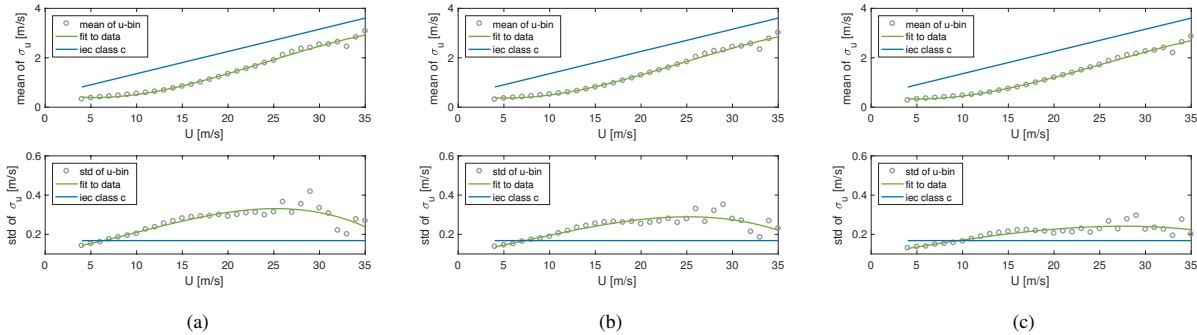

**Figure A1.** Notation same as Figure 4 but for a) linearly detrended data, b) high-pass filtered data with cut-off frequency of 1/600 Hz and c) high-pass filtered data with cut-off frequency of 1/300 Hz.

## Appendix A

The Figure in this appendix is equivalent to Figure 4, but shows the processed measurements.

Comparing the raw data in Figure 4, to the linearly detrended data and high-pass filtered data in Figure A1 it is seen that the detrending, and high pass filtering slightly lowers the values of $\mu_{\sigma_u}$, while the reduction of $\sigma_{\sigma_u}$ is much greater, especially for the high-pass filtered measurements.

## Appendix B

Figure B1 shows extreme moments as function of the u-component of the mean hub-height wind speed. Each dot shows the maximum/minimum load value of each 10-minute HAWC2 simulation for the tower top (top panels), the tower base (middle panels) and blade root (bottom panels). The simulations based on a particular extreme variance event may be identified as a cluster of six dots, as they have been simulated with six different turbulence seeds. For DLC 1.3 a cluster of six dots may be seen, as the simulations are performed with six turbulence seeds per mean wind speed step. Figure 9 shows the values from Figure B1, binned and averaged.

*Author contributions.* ÁH performed the data analysis and simulations. ÁH made all figures. MK provided guidance and comments. ND developed the code that is used to perform constrained turbulence simulations. ÁH prepared the manuscript with contributions from the co-authors. This work is part of ÁH's PhD under supervision of MK.

*Competing interests.* The authors declare that no competing interests are present in this work.

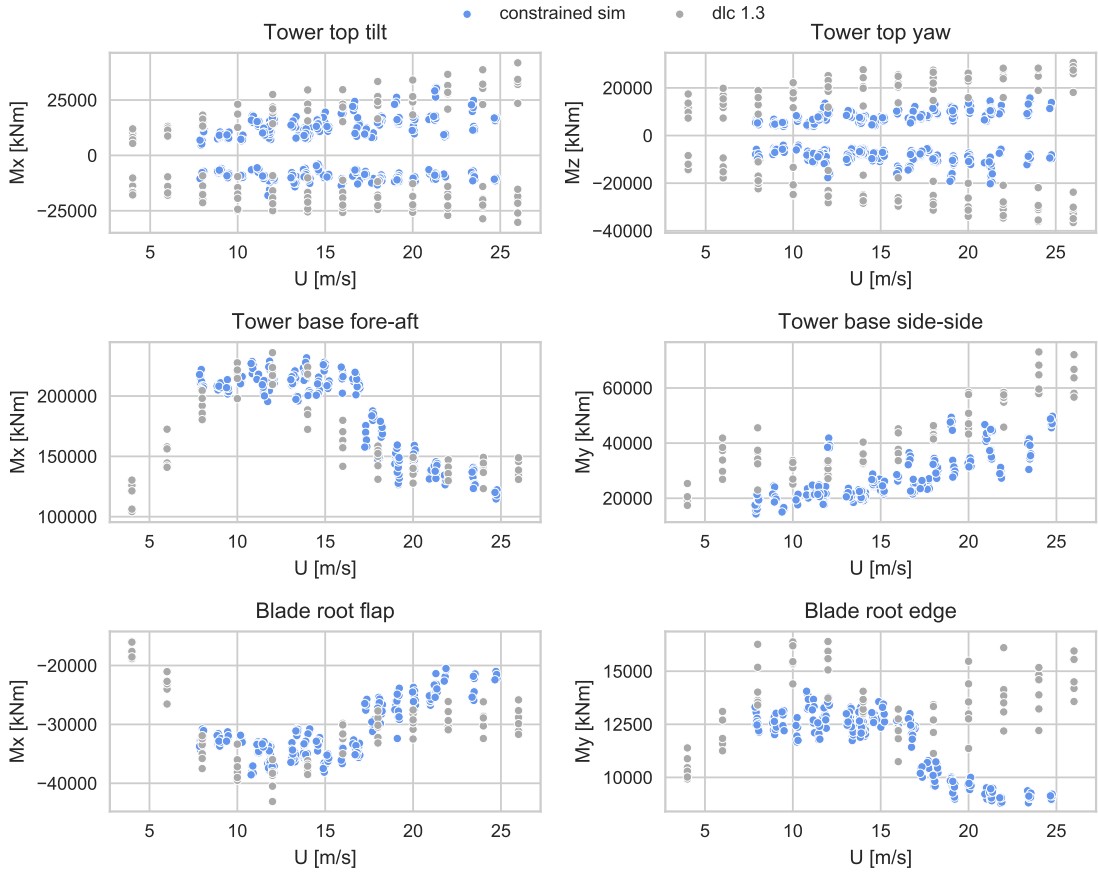

**Figure B1.** The extreme moments from IEC DLC 1.3 (grey dots). The extreme loads from the constrained simulations (blue dots).

*Acknowledgements.* The authors would like to thank Anand Natarajan and Jakob Mann for constructive comments and discussion. ÁH would also like to acknowledge Jenni Rinker and David Verelst for HAWC2 assistance.

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
