# Peer review of "Extreme wind fluctuations: joint statistics, extreme turbulence, and impact on wind turbine loads"

_Wind Energy Science, 2018_

## Referee Comment (RC1) · Anonymous Referee #1 · 20 Apr 2018

**Review of Extreme fluctuations of wind speed for a coastal/offshore climate: statistics and impact on wind turbine loads**
*Hannesdóttir, Kelly, Dimitrov*

The manuscript considers extreme fluctuations via a turbulence model per IEC to assess loadings on turbines. The model follows data taken over the coast of Denmark. The manuscript is motivated using arguments as proposed in standards for generating the fields and then observing their influence on the blade and tower relating to the

various moments associated. The topic is of interest, by and large, to the wind energy and atmospheric science community. The manuscript provides justification for assumptions taken in almost its entirety, which is seen as positive. The manuscript is generally well written and its results substantiated by data. The manuscript would benefit by considering the points below.

0.) The title should be modified to more accurately represent the content of the manuscript; 1.) Including salient results in the abstract; 2.) Reducing non-descriptive adjectives in the introduction (big, short, etc.); 3.) Providing further detail on the site and measurements as these are critical to the overall framing of the manuscript; 4.) In figure 3 and 4, for example, subfigures are not discussed in their entirety - if not discussed then these should be removed; 5.) Placement of figures tend to occur prior to the narration; 6.) Comment on process for figure 4 to go from raw measurements to high-pass filtered measurements more carefully; 7.) When discussing design load cases and simulations, consider non-Gaussian fields as it is known that realistic fields may differ from Gaussian; 8.) Include literature on works considering conditional pdfs in regards to turbulence fields/statistics/wind power; 9.) Content/results on p12 should be expanded - this is the case with most results that physics based observations are missing. In this case, given that the simulations are based on a model, it is relevant to justify their physicality; 10.) §5 is difficult to follow and should be revisited as well as explaining the results in more detail; 11.) Figure quality may be improved; 12.) Conclusions can be presented in non-bullet form and at the present the discussion and conclusions sections may be combined; 13.) Is it possible to extend the analysis to further cases for sake of comparison?

---

## Referee Comment (RC2) · Anonymous Referee #2 · 16 May 2018

This paper contains significant work that can assist in updating the Extreme Turbulence Model (ETM) of IE61400-1 in order to improve the prediction of extreme tower base fore-aft loads in the extreme design load case 1.3. There are a number of researchers connected with the IEC 61400 series maintenance teams as well as the IEA Wind R&D groups who think that the extreme wind condition modelling in the 61400 series does not reflect the kind of extreme wind events that occur in nature. This work is promising, particular if it is extended to consider other extreme design load cases such as EOG, ECD and EWS.

The scientific approach appears valid. I did wonder why TI was used to isolate the extreme variance events though. Why not just look at a plot of wind speed standard deviation versus wind speed? I also was not clear about the process of excluding measurements from the wake of nearby wind turbine. Was this exclusion of sectors covering 0 -180 degrees?

Presentation is very good in general. I have uploaded an annotated pdf with comments that may help to improve clarity. For instance, I think that the caption Figure 6 should refer to z = 119 m since line 10 on page 12 mentions the time series are at hub-height.

Please also note the supplement to this comment:
https://www.wind-energ-sci-discuss.net/wes-2018-12/wes-2018-12-RC2-supplement.pdf

**Supplement:**

[revised manuscript text omitted]

---

## Author Comment (AC1) · 14 Jul 2018

**1    Review by anonymous Referee 1**

*The manuscript considers extreme fluctuations via a turbulence model per IEC to assess loadings on turbines. The model follows data taken over the coast of Denmark. The manuscript is motivated using arguments as proposed in standards for generating the fields and then observing their influence on the blade and tower relating to the various moments associated.   The topic is of interest, by and large, to the wind energy and atmospheric science community. The manuscript provides justification for*

[Figure]

*assumptions taken in almost its entirety, which is seen as positive. The manuscript is generally well written and its results substantiated by data. The manuscript would benefit by considering the points below.*

*0.) The title should be modified to more accurately represent the content of the manuscript;*

*1.) Including salient results in the abstract;*

*2.) Reducing non-descriptive adjectives in the introduction (big, short, etc.);*

*3.) Providing further detail on the site and measurements as these are critical to the overall framing of the manuscript;*

*4.) In figure 3 and 4, for example, subfigures are not discussed in their entirety - if not discussed then these should be removed;*

*5.) Placement of figures tend to occur prior to the narration;*

*6.) Comment on process for figure 4 to go from raw measurements to high-pass filtered measurements more carefully;*

*7.) When discussing design load cases and simulations, consider non-Gaussian fields as it is known that realistic fields may differ from Gaussian;*

*8.) Include literature on works considering conditional pdfs in regards to turbulence fields/statistics/wind power;*

*9.) Content/results on p12 should be expanded - this is the case with most results that physics based observations are missing. In this case, given that the simulations are based on a model, it is relevant to justify their physicality;*

*10.) §5 is difficult to follow and should be revisited as well as explaining the results in more detail;*

*11.) Figure quality may be improved;*

*12.) Conclusions can be presented in non-bullet form and at the present the discussion and conclusions sections may be combined;*

*13.) Is it possible to extend the analysis to further cases for sake of comparison?*

Reply to reviewer 1:

Thank you for very much for your constructive review and comments. We have considered your suggestions and made changes according to them. In the following we show our response in the same order as the comments:

0) We agree with you and have changed the title of the paper to: Extreme wind fluctuations: joint statistics, extreme turbulence, and impact on wind turbine loads

1) More details of the results have been added to the abstract.

2) Adjectives have now been removed from the introduction.

3) We have added a figure showing the location and a overview of the measurement site, also more text.

4) In Figure 3, the subfigures are mentioned on page 6, line 16, however it previously was not very clear. Thus we agree that it is not necessary to include all the subfigures, as they are so similar. We have moved the subfigures in Figure 3 to an appendix, so they can still be viewed by a reader interested in those details. For Figure 4 the explicit mention of each subfigure has been made clearer.

5) This was due to the WES latex package and recommendation, but we have manipulated it somewhat to show the figures closer to the corresponding text.

6) We think this is a good suggestion, and we have added the expression for the frequency response function of the Butterworth filter that we use. We also changed the the the cut-off frequency for the high-pass filtering to be more conservative, and added a subfigure where the data is filtered with even lower cut-off frequency. The measurements are high-pass filtered with cut-off frequencies of 1/600 Hz and 1/300 Hz, instead of 1/200 Hz only.

7) In §4.3 we have added a consideration on non-Gaussian fields and state that the difference between Gaussian and non-Gaussian turbulence as input to load simulations has been shown to give insignificant difference in load results. This has been shown by Berg et al (2016).

8) We have added three references: Fitzwater et al.(2003), Saranyarsoontorn and Manuel (2006) and Moon et al. (2014) in §3.1

9) As you say, we do not have physical observations of wind turbine loads during

these extreme variance events. This is why we simulate them in HAWC2 which is a model. This model has been referenced in §4.1, but we have included a more in-depth explanation of the HAWC2 model.

10) Yes, §5 has been polished for a hopefully easier flow. Figure 8 have been removed (moved to appendix B), as Figure 9 shows the binned and average values of Figure 8. The text is trimmed. The main points should now stand out more clearly, while they have also been expanded on.

11) Figures 1,3-5 and 7-11 now have a higher resolution.

12) Yes, we have considered a more conventional conclusion section, without bullets. The purpose of the bullets is to give the reader a quick overview and an easier focus on the main findings, and after trying both versions we have decided to keep the bullets.

13) Yes, it is definitely possible for future work, e.g including more measurement sites or by lowering the curve of the selection criteria for the present site.

Note: In the edited version of the manuscript the figure numbering differs from the original one, due to adding/removing of plots. In the response we refer to the original manuscript figure numbering.

**2 Review by anonymous Referee 2**

*This paper contains significant work that can assist in updating the Extreme Turbulence Model (ETM) of IE61400-1 in order to improve the prediction of extreme tower base fore-aft loads in the extreme design load case 1.3. There are a number of researchers connected with the IEC 61400 series maintenance teams as well as the IEA Wind R&D groups who think that the extreme wind condition modelling in the 61400 series does not reflect the kind of extreme wind events that occur in nature. This work is promising,*

*particular if it is extended to consider other extreme design load cases such as EOG, ECD and EWS.*

*The scientific approach appears valid. I did wonder why TI was used to isolate the extreme variance events though. Why not just look at a plot of wind speed standard deviation versus wind speed? I also was not clear about the process of excluding measurements from the wake of nearby wind turbine. Was this exclusion of sectors covering 0 -180 degrees?*

*Presentation is very good in general. I have uploaded an annotated pdf with comments that may help to improve clarity. For instance, I think that the caption Figure 6 should refer to z = 119 m since line 10 on page 12 mentions the time series are at hub-height.*

*Please also note the supplement to this comment: https://www.wind-energ-sci-discuss.net/wes-2018-12/wes-2018-12-RC2-supplement.pdf*

Reply to reviewer 2: Thank you for the positive comments and constructive suggestions. We have used some of your language usage and phrasing suggestions (where appropriate) and made changes accordingly. We reply to your comments in the same order as they appear the annotated pdf-file, disregarding usage/phrasing comments:

*"perhaps need to expand as per the abstract - The variance of wind velocity fluctuations manifested during these events is not due to extreme turbulence; rather, it is primarily caused by ramp-like increases in wind speed associated with larger-scale meteorological processes."*

Answer: We have expanded the text in the introduction as you suggest.

*"lighting mast? instrumented with lights for the test site? or lightning mast? with lightning rods to protect the turbines?"*

Answer: It is a light mast, with aircraft warning lights on the top. We have added a footnote explaining this.

*" I presume wind speeds from the meteorological mast are used to correlate with power from the wind turbines and that is why is was important to compare lighting mast and main met mast (if you are later going to look at wind turbine performance)."*

Answer: The comparison of the met-mast data and the light-mast data is made to demonstrate that the extreme events are large coherent structures, as seen in Figure 2 and discussed in the text.

*"would be nice to have a figure here to show the site layout."*

Answer: We think this is an excellent suggestion and we have added a map of Høvsøre and an overview of the site.

*"can this Figure be sited closer to the reference to the Figure in the text?"*

Answer: This was due to the WES latex package and recommendation, but we have manipulated it somewhat to show the figures closer to the corresponding text.

*"why not just look at sigma - the 10-minute standard deviations to find the extreme variance events? Low wind speeds may give misleading high TI values."*

Answer: We agree with you and we have changed Figure 1 to show10-minute standard deviations as function of 10-minute mean wind speed, instead of TI vs U.

*"clarify here extreme turbulence model is a function all of the aforementioned parameters in the sentence??"*

[Figure]

Answer: We mean that $\sigma_1$ is a linear function of hub-height wind speed (following the IEC 61400-1). It could be written as $\sigma_1(V_{hub})$ in the standard, i.e. $V_{hub}$ is the variable and other parameters are constants. We have clarified in the text.

*"again could not low wind speed values influence the TI results. Can you say that the blue dots above the blue curve are events with high variance?"*

Answer: It is now more clear as Figure 1 has been changed to: sigma vs U, according to your suggestion.

*"again - helpful to have figure 3 closer to the text in which it is referred to. I assume this will be sorted out during publishing."*

Answer: Has been modified.

*"consider using ,respectively at the end of the sentence"*

Answer: Has been changed according to your suggestion.

*"fluctuated below 180 deg? i.e. including small fluctuations in wind direction? Is this just discounting certain sectors? What is meant by below?"*

Answer: In a few cases the wind direction changed so it was temporarily from South (180 deg), while the mean direction was still from West. The sentence has been changed to: Finally, events where the corresponding directional data fluctuated below 180° are discarded, i.e. temporary directional data from South, to exclude measurements from the wake of the nearby wind turbine.

*"an illustrative diagram would be useful here"*

Answer: We believe that Figure 5 serves to show the dimensions of the turbulence boxes.

*"rotor speeds seem very low"*

Answer: It does, but this is the correct value. At 9.6 rpm the tip speed of the blade is 90 m/s for the DTU 10 MW. For comparison the NREL 5 MW has a rated rotor speed of 12.1 rpm and maximum tip speed of 80 m/s.

*"are these corresponding to the six events as shown in Figure 2?"*

Answer: No, they are randomly synthesized seeds. The word synthesized has been added to the stance to clarify.

*"source time series is perhaps a bit confusing - may be interpreted as measured time series?"*

Answer: We think you are right, and have changed the legend in Figure 6 to say instead: Synthesized time series.

*"if at hub-height ,then z =119m"*

Answer: Thank you for pointing this out. This has now been changed.

*"The layout of the figure could be made clearer. e.g. mark underneath*
*(a) DLC 1.3*
*(b) constrained*
*top panels - wind speed*
*bottom panels - moments"*

[Figure]

Answer: They layout of the figure has been changed to make clearer.

**Supplement:**

[revised manuscript text omitted]

---

## Author Response (AR2)

Comments from reviewer #1:
*The manuscript now contains most suggestions as presented in the first iteration of the revision. One outstanding comment is to consider referencing works by Peinke, Schmitt, Cal, Chamorro and Wilczek.*

Answer: We have now included relevant references of Peinke et al. (2004), Wilczek and Friedrich (2009) and Chamorro et al. (2015). Also a reference with the work of Morales et al. (2012) was added.

Comments from reviewer #3:
*As I will contribute only to the second round of reviews, and most issues have already been addressed, the review will be restricted to remaining issues.*

*The paper reports on work about an interesting and relevant topic, and is expected to receive considerable interest. The scientific work is in general sound and thorough, and well presented.*

*There are two remaining issues, which follow two recommendations of referee 1.*

*First, as stated by referee 1 in issue 9.), the consideration of realistic, non-Gaussian turbulence would improve the results. The paper justifies the usage of Gaussian turbulence fields by citing Berg et al. (2016), namely their conclusion that difference to non-Gaussian turbulence with respect to wind turbine loads would be insignificant. This cannot be accepted.*
*While this special conclusion is claimed in the cited paper, the contents and results of Berg et al. (2016) do not support that claim. Moreover, other publications have clearly shown the relevant influence of intermittency, such as [1,2]. There may be, however, many other possible reasons to restrict the investigations to Gaussian turbulence, such as restricting the complexity of the set-up, availability of widely accepted codes, numerical efficiency, etc.*

Answer:
A further discussion on Gaussian and non-Gaussian turbulence fields has been added to the discussion (section 6) and the suggested references have been added (Mücke et al. (2011) and Schottler et al. (2017))

*Second, similar to what was stated by referee 1 in issue 8.), a deeper analysis of the measured wind speed time series would be beneficial, namely in terms of higher-order two-point statistics. This can be done by conditional pdfs p(u(t+tau)|u(t)) as suggested by referee 1, or by increment pdfs p(u(t+tau)-u(t)). The relevant time scales tau are those where there is dynamic response of the turbine to turbulence, typically in the range of 1 to 100 s. Relevant literature should be available. The inclusion of the three references to works by Fitzwater, Moon, and Saranyasoontorn (while still helpful for the paper) does not seem to answer the recommendation of referee 1.*

*With these two issues addressed, I would consider the paper to be ready for publication.*

*[1] Schottler et al., Wind Energ. Sci. 2, 1-13, 2017*
*[2] Mücke et al., Wind Energ. 14, 301-316, 2011*

Answer:
As suggested by referee #1 in point 8 we have included literature on works considering conditional pdfs in regards to turbulence fields/statistics/wind power and now added more references.
We believe that including two-point increment statistics (conditional or not) of the set of highly non-stationary time series considered here is out of the scope of the paper. We believe this because the events correspond to inhomogeneous (non-turbulent) field fluctuations. An increment study of non-stationary time series might result in misleading intermittent distributions, that are due to the change in mean wind speed (large scale weather phenomena), rather than intermittent turbulence. An increment analysis of the analysed events requires great care and a number of further considerations, perhaps deserving a publication of its own. The increment studies in the added references also clearly state the importance of stationarity for such analysis (Mücke et al. (2011), Schottler et al. (2017), Morales et al. (2012)).

[revised manuscript text omitted]